# L-Ascorbate Biosynthesis Involves Carbon Skeleton Rearrangement in the Nematode *Caenorhabditis elegans*

**DOI:** 10.3390/metabo10080334

**Published:** 2020-08-17

**Authors:** Yukinori Yabuta, Ryuta Nagata, Yuka Aoki, Ayumi Kariya, Kousuke Wada, Ayako Yanagimoto, Hiroka Hara, Tomohiro Bito, Naho Okamoto, Shinichi Yoshida, Atsushi Ishihara, Fumio Watanabe

**Affiliations:** 1Department of Agricultural, Life and Environmental Sciences, Faculty of Agriculture, Tottori University, 4-101 Koyama-Minami, Tottori 680-8553, Japan; b17a5143u@edu.tottori-u.ac.jp (R.N.); koromimi.love.0703@gmail.com (Y.A.); kariyaayumi11@gmail.com (A.K.); vitaminc198.11@gmail.com (K.W.); ascorbate250@gmail.com (A.Y.); a3416341@yahoo.co.jp (H.H.); bito@tottori-u.ac.jp (T.B.); aishihara@tottori-u.ac.jp (A.I.); watanabe@tottori-u.ac.jp (F.W.); 2The United Graduate School of Agricultural Sciences, Tottori University, 4-101 Koyama-Minami, Tottori 680-8553, Japan; d18a2005y@edu.tottoir-u.ac.jp; 3Electronic and Organic Material Laboratory, Tottori Institute of Industrial Technology, 7-1-1 Wakabadai-minami, Tottori 689-1112, Japan; syoshida@tiit.or.jp

**Keywords:** L-ascorbate (AsA), antioxidant, biosynthetic pathway, *Caenorhabditis elegans*, reactive oxygen species, redox

## Abstract

Ascorbate (AsA) is required as a cofactor and is widely distributed in plants and animals. Recently, it has been suggested that the nematode *Caenorhabditis elegans* also synthesizes AsA. However, its biosynthetic pathway is still unknown. To further understand AsA biosynthesis in *C. elegans*, we analyzed the incorporation of the ^13^C atom into AsA using gas chromatography-mass spectrometry (GC-MS) in worms fed with D-Glc (1-^13^C)-labeled *Escherichia coli*. GC-MS analysis revealed that AsA biosynthesis in *C. elegans*, similarly to that in mammalian systems, involves carbon skeleton rearrangement. The addition of L-gulono-1,4-lactone, an AsA precursor in the mammalian pathway, significantly increased AsA level in *C. elegans*, whereas the addition of L-galactono-1,4-lactone, an AsA precursor in the plant and *Euglena* pathway, did not affect AsA level. The suppression of E03H4.3 (an ortholog of gluconolactonase) or the deficiency of F54D5.12 (an ortholog of L-gulono-1,4-lactone oxidase) significantly decreased AsA level in *C. elegans*. Although N2- and AsA-deficient F54D5.12 knockout mutant worm (tm6671) morphologies and the ratio of collagen to non-collagen protein did not show any significant differences, the mutant worms exhibited increased malondialdehyde levels and reduced lifespan compared with the N2 worms. In conclusion, our findings indicate that the AsA biosynthetic pathway is similar in *C. elegans* and mammals.

## 1. Introduction

L-Ascorbate (AsA), also known as vitamin C, is a typical antioxidant with an important role in the biosynthesis of collagen, carnitine, and neurotransmitters [1]. Three enzymes, prolyl-3-hydroxylase, prolyl-4-hydroxylase, and lysyl hydroxylase, are central to collagen biosynthesis and are responsible for the production of 3-hydroxyproline, 4-hydroxyproline, and hydroxylysine, respectively [2]. These enzymes require reduced iron (Fe^2+^) and AsA as a reducing agent for their activities. The 4-hydroxyproline residues are essential for the stability of the collagen triple helix [3]. The hydroxylysine residues of the collagen molecule are glycosylated by galactose or glucose-galactose addition [4]. Such glycosylation is key for collagen interaction with certain receptors [5,6,7,8]. Although the molecular function of 3-hydroxyprolines has been elusive, Hudson et al. [9] have reported that type I collagen 3-hydroxyproline residues might contribute to fibril supramolecular assembly.

ε-N-trimethyl-L-lysine hydroxylase and γ-butyrobetaine hydroxylase are involved in carnitine biosynthesis, which is essential for mitochondrial fatty acid transport [10,11]. These enzymes also require Fe^2+^ and AsA for their activities. Dopamine ß-hydroxylase catalyzes dopamine conversion into norepinephrine, and its activity requires AsA as a cofactor for copper reduction [12].

Plants, including eukaryotic algae, and most animals except for primates, guinea pigs, bats, and some birds, synthesize AsA. Primates, including humans and guinea pigs, thus depend on dietary AsA intake to prevent scurvy. In mammals, AsA is synthesized in the liver or kidney from D-glucose (D-Glc) through D-glucuronic acid, L-gulonic acid and L-gulono-1,4-lactone (Figure 1) [13]. SMP30 (senescence marker protein 30)/gluconolactonase (GNL) catalyzes L-gulonic acid conversion into L-gulono-1,4-lactone [14]. SMP30 expression reportedly decreases with aging in rat livers, kidneys, and lungs [15,16]. SMP30 knockout mice exhibit significantly lower AsA levels in their plasma, livers, and kidneys than wild-type mice [14]. The final step of AsA biosynthesis is microsomal L-gulono-1,4-lactone oxidase (GULO), which catalyzes L-gulono-1,4-lactone conversion into AsA [17]. The inability of primates and guinea pigs to synthesize AsA is due to the lack of GULO. Interestingly, guinea pig and human GULO genes are highly mutated and do not encode a functional protein [18,19]. In this pathway, D-Glc is oxidized at C6, forming D-glucuronic acid, which undergoes reduction at C1, forming L-gulonic acid. Accordingly, the C1 of D-Glc becomes the C6 of AsA. Hence, AsA biosynthesis involves carbon skeleton rearrangement in mammals.

Plants synthesize AsA from D-Glc through D-Glc-6-phosphate, D-fructose-6-phosphate, D-mannose-6-phosphate (D-Man-6-phosphate), D-Man-1-phosphate, GDP-D-Man, GDP-L-galactose, L-galactose-1-phosphate, L-galactose, and L-galactono-1,4-lactone (Figure 1). This pathway is called the Smirnoff–Wheeler pathway or the D-mannose/L-galactose pathway and is supported by genetic evidence from *Arabidopsis thaliana* [20,21]. This pathway does not involve carbon skeleton rearrangement. Although alternative pathways were found in higher plants [22,23,24], the evidence for their existence is largely based on increased AsA levels induced by ectopic gene expression. Hence, the Smirnoff–Wheeler pathway is the primary AsA biosynthesis pathway. Mitochondrial L-galactono-1,4-lactone dehydrogenase (L-GalLDH) catalyzes L-galactono-1,4-lactone conversion into AsA in both pathways [25,26,27].

*Euglena gracilis* contain significant concentrations of AsA. In *Euglena*, AsA is synthesized from D-Glc through UDP-D-glucuronate, UDP-D-galacturonate, D-galacturonate, and L-galactonate (Figure 1) [28]. We reported that SMP30 homolog aldonolactonase catalyzes L-galactonate conversion into L-galactono-1,4-lactone, and that AsA biosynthesis predominantly imparts the D-galacturonate/L-galactonate pathway in *Euglena* [29]. The final step is catalyzed by L-GalLDH. Similarly to the mammalian pathway, the AsA biosynthesis in *Euglena* also accompanies the rearrangement of the carbon skeleton [21,28].

The nematode *Caenorhabditis elegans* expresses a wide range of proteins that belong to the collagen family, including cuticle collagens with nearly 175 different polypeptides and two basement membrane collagens [3]. Mutations in the cuticle collagen genes reportedly affect body morphology. Furthermore, Winter and Page [30] reported that prolyl 4-hydroxylase is required for exoskeleton formation and the maintenance of body shape in *C. elegans*.

The mammalian prolyl 4-hydroxylase is a hetero-tetramer (α2/β2), consisting of a catalytic α and a disulfide-isomerase (PDI)-homolog β subunit. In *C. elegans*, the α subunits are encoded by four genes (*phy-1* to *phy-4* [3]), whereas PDI has two isoforms, PDI-1 and PDI-2. PDI-2 is the prolyl 4-hydroxylase β subunit involved in cuticle collagen synthesis. Veijola et al. [31] reported that the hybrid prolyl 4-hydroxylase, which consists of the *C. elegans* α subunit (PHY-1) and human PDI/β subunit, utilizes AsA. These findings strongly indicate that *C. elegans*, like mammals, requires AsA for its normal development.

It has been recently reported that the nematode *C. elegans* can synthesize AsA. ^13^C-labeled AsA was detected in *C. elegans* when the worms were fed *Escherichia coli* grown in a ^13^C-labeled D-Glc (U-^13^C_6_)-containing medium [32]. However, the biosynthetic pathway is still unknown. As described above, the difference between the mammalian, plant, and *Euglena* AsA biosynthesis is the presence or absence of carbon skeleton rearrangement. Focusing on this difference, we explored the AsA biosynthetic pathway in *C. elegans* using gas chromatography-mass spectrometry (GC-MS). To the best of our knowledge, this is the first study that identifies the AsA biosynthetic pathway in *C. elegans*.

## 2. Results

### 2.1. AsA Biosynthesis in C. elegans Involves Carbon Skeleton Rearrangement

We analyzed AsA to validate its presence in unlabeled *E. coli*-fed *C. elegans* using GC-MS. Since the BSTFA/TMCS derivatized standard AsA retention time was 8.1 min (Appendix A), we analyzed the mass spectrum. The ion at *m*/*z* 464 was attributed to the trimethylsilylated-AsA precursor ion, whereas its major fragment ions were detected at *m*/*z* 449, *m*/*z* 332, *m*/*z* 304, *m*/*z* 259, *m*/*z* 205, and *m*/*z* 117 (Appendix A). This mass spectrum corresponds to that of trimethylsilylated-AsA. The ions at *m*/*z* 449, *m*/*z* 332, *m*/*z* 304, *m*/*z* 259, *m*/*z* 205, *m*/*z* 147, and *m*/*z* 117 reportedly contained six, four, four, four, two, two, and two carbon atoms of the derivatized AsA, respectively [32]. When *C. elegans* were fed with D-Glc (U-^13^C_6_)-labeled *E. coli*, the ions at *m*/*z* 455, *m*/*z* 336, *m*/*z* 307, *m*/*z* 263, *m*/*z* 207, and *m*/*z* 119 were detected as derivatized AsA fragments in the extract (Appendix A), indicating that *C. elegans* can synthesize AsA. This finding was consistent with previous reports [32]. Similar to a previous report [32], there was no observed ^13^C atom incorporation into the 147 *m*/*z* fragment ion, suggesting the possibility that this ion does not derive from AsA.

The ions at *m*/*z* 465 and *m*/*z* 450 were detected in D-Glc (1-^13^C)-labeled *E. coli*-fed *C. elegans* (Figure 2), indicating that these AsA-derived ions were synthesized from D-Glc (1-^13^C). The ions *m*/*z* 118 and *m*/*z* 206 were detected at higher strength, compared with those in unlabeled *E. coli*-fed *C. elegans*. The ions at *m*/*z* 117 and *m*/*z* 205 contained the C6 of AsA [32]. To prepare AsA (1-^13^C), spinach leaves were supplemented with 5 mM D-Man (1-^13^C) for 24 h. Although the ions at *m*/*z* 465 and *m*/*z* 450 in the labeled leaves were detected at higher strength, compared with those in the non-labeled D-Man-supplemented leaves (non-labeled leaves), no significant difference was observed in the ion strength at *m*/*z* 205 and *m*/*z* 117 between the labeled and non-labeled leaves (Appendix A). These findings indicate that AsA biosynthesis in *C. elegans* involves carbon skeleton rearrangement.

### 2.2. Addition of L-Gulono-1,4-lactone or L-Galactono-1,4-lactone, AsA Precursors, to C. elegans

AsA biosynthesis in mammals or *Euglena* involves in the rearrangement of the carbon skeleton. GC-MS analysis clearly indicated that the AsA biosynthetic pathway in *C. elegans* is similar to the mammalian or *Euglena* pathways. The AsA precursor in mammals and *Euglena* are L-gulono-1,4-lactone and L-galactono-1,4-lactone, respectively. To determine whether worms are using either of these precursors, we performed feeding experiments.

In those worms that were fed with L-gulono-1,4-lactone-packed liposomes, AsA contents were approximately 1.43-fold higher than in those fed with the empty liposomes (Figure 3). No significant difference was observed in the AsA contents between the control worms and those that were fed with L-gulono-1,4-lactone-packed liposomes. These findings suggest that the AsA biosynthetic pathway in *C. elegans* was similar to the mammalian pathway.

### 2.3. Identification of the AsA Biosynthetic Pathway in C. elegans Using Knockout or Knockdown Worms

To identify the AsA biosynthetic pathway in *C. elegans*, we searched for such protein-coding genes from the *C. elegans* genome that show homology with rat SMP30/GNL (National Center for Biotechnology Information (NCBI) protein ID; NP_113734) or GULO (NCBI protein ID; NP_071556) using the WormBase database [33]. We found that E03H4.3 and F54D5.12 were similar to SMP30/GNL and GULO, respectively. The deduced amino acid sequences of E03H4.3 showed 28% identity and 70% similarity with rat SMP30/GNL, respectively (Appendix A). F54D5.12 shared 27% identity and 68% similarity with rat GULO, respectively (Appendix A).

We managed to obtain F54D5.12 knockout (tm6671) worms. However, not only the first exon of the F54D5.12 gene, but also a part of the 3′- untranslated region of the F54D5.11 gene were deleted in the tm6671 mutant worms. We considered the possibility that the transcript level of F54D5.11 was changed in the tm6671 mutant worms. Therefore, we checked the transcript level of F54D5.11 in the N2 and tm6671 mutant worms. No significant difference was observed in the transcript level of F54D5.11 between the N2 and tm6671 mutant worms (Appendix A). It seems likely that the expression of the F54D5.11 protein is not affected by the tm6671 allele.

Since we could not obtain any E03H4.3 knockout mutant strain, we downregulated E03H4.3 expression using RNA interference (RNAi)-knockdown. The AsA content in the E03H4.3 knockdown worms was lower (82.2%) than in the control worms (KP3948) (Figure 4). In tm6671 mutants, AsA was significantly lower (81.1%) than in N2 worms (Figure 4). Thus, these findings clearly showed that the deficiency of E03H4.3 or F54D5.12 expression decrease the AsA content in the worms.

Furthermore, we found that the deduced Y50D7A.7 amino acid sequences showed 30% identity and 68% similarity with *Euglena* L-GalLDH (accession number, AB914478), respectively. Therefore, we performed RNAi experiments. Y50D7A.7 downregulation did not affect the AsA contents (Appendix A). The BX10 (*wa3*) strain carries a point mutation at position 1361 in the Y50D7A.7 open reeding frame (ORF) encoding a Gly-to-Asp change at position 454 of the translated sequence [34]. Interestingly, no significant difference was observed between the N2 and mutant worms in the AsA contents (Appendix A).

### 2.4. Reduced AsA Levels Affect C. elegans Lifespan and Lipid Peroxide Levels

We also investigated the effect of reduced AsA levels in worms. The ratio of collagen to non-collagen protein was measured in the N2 and tm6671 mutant worms but did not show any significant difference (Appendix A). Furthermore, we could not observe any difference in morphology between the N2 and tm6671 mutant worms (Appendix A). Interestingly, the lifespan of tm6671 mutant worms was significantly reduced, compared with that of the N2 worms (Figure 5). The maximal lifespan of tm6671 mutant worms was 18 days, compared with a lifespan of 21 days in the N2 worms. It seems likely that oxidative stress was induced in the tm6671 mutant worms. To evaluate the oxidative stress levels, we measured the levels of malondialdehyde (MDA), a product of lipid peroxidation. In the tm6671 mutant worms, the MDA level was 1.75-fold higher than in the N2 worms (Figure 6).

## 3. Discussion

Our present study has revealed that the AsA biosynthetic pathway in *C. elegans* involves carbon skeleton rearrangement through L-gulono-1,4-lactone, and the *C. elegans* and mammalian AsA biosynthetic pathways are thus similar. Although the deficiency of F54D5.12, which is a homolog with that of rat GULO, decreased AsA level, the rate of decline in mutant worms was only 18.9% (Figure 4), suggesting that other proteins might compensate for the role of F54D5.12. We found that Y7A5A.1 and F52H2.6 were homologous to rat GULO. To examine the involvement of these proteins in AsA biosynthesis, we are going to analyze how the reduced Y7A5A.1 or F52H2.6 expression levels affect AsA level.

Mutations in cuticle collagens and their modifying enzymes affect worm body morphology [3]. *C. elegans* has a single lysyl hydroxylase gene (*let*-268) expressed in type IV collagen-producing cells [35]. The *let-268* mutation impairs type IV collagen secretion into the basement membrane and results in embryonic lethality. The *phy-1* mutation causes a dumpy phenotype, whereas the *phy*-*2* mutation does not affect morphology [3,30,36,37,38]. *phy-1*/*phy-2* double mutants and *pdi-2* null mutants are embryonic lethal [3,30,36], whereas *phy-3* mutants have a wild-type phenotype [39] and the *phy-4* mutant phenotype remains unclear. Interestingly, an approximately 20% AsA level reduction does not affect collagen level and morphology in tm6671 worms.

The lifespan of tm6671 mutant worms with reduced AsA levels was significantly shorter than those of N2 worms (Figure 5). Because the MDA level in tm6671 mutant worms was significantly higher, compared with that in the N2 worms (Figure 6), the enhanced oxidative stress seemed to reduce tm6671 worm lifespan. It is well known that oxidative stress reduces the lifespan of *C. elegans* [40,41]. These findings suggest that enhanced oxidative stress shortens lifespan in tm6671 worms.

Univalent AsA oxidation leads to the formation of monodehydro-AsA (MDHA) that is converted into the divalent oxidation product dehydro-AsA (DHA) through spontaneous disproportionation or further oxidation. MDHA and DHA are reduced into AsA by non-enzymatic and enzymatic reactions. Enzymatic reactions involve MDHA and DHA reductase in animals and plants, respectively. However, no specific MDHA reductase proteins have yet been characterized in animals [42]. MDHA was reportedly reduced by NADH-cytochrome *b_5_* [43] and thioredoxin reductase in rat livers [44]. As DHA is unstable at physiological pH and temperature [45,46], AsA regeneration from DHA could be a beneficial process [47]. DHA reduction is catalyzed by several enzymes, such as omega class glutathione *S*-transferase, glutaredoxin, and PDI with GSH-dependent DHA reductase activity in mammals [48,49,50]. 3α-Hydroxysteroid dehydrogenase and thioredoxin reductase exhibit NADPH-dependent DHA reductase activity [51,52]. The homologs of these genes are conserved in *C. elegans*.

In conclusion, our study proves that AsA biosynthesis in *C. elegans* involves the same carbon skeleton rearrangement as that in mammals. 

## 4. Materials and Methods 

### 4.1. Chemicals

D-Glc (1-^13^C), D-Glc (U-^13^C_6_) and D-Man (1-^13^C) were purchased from Cambridge Isotope Laboratories (Tewksbury MA, USA). A mixture of *N*,*O*-bis(trimethylsilyl)trifluoroacetamide/trimethylchlorosilane (BSTFA/TMCS) (99:1, *v*/*v*) was purchased from Supelco, Bellefonte (PA, USA). 1,2-Dioleoyl-sn-glycero-3-phosphocholine (DOPC) was purchased from Avanti polar lipids (Alabaster, AL). L-galactono-1,4-lactone was purchased from Carbosynth Ltd. (Newbury, Berkshire, UK). All other chemicals were of an analytical grade and were purchased from a commercial source.

### 4.2. Worm Cultures and Strains

The N2 Bristol *C. elegans* strain was maintained at 20 °C on nematode growth medium (NGM) plates using the *Escherichia coli* OP50 strain as a food source [53]. The mutant strain was backcrossed five times before further analysis. Nematode synchronization was performed using the standard alkaline hypochlorite method.

### 4.3. Preparation of ^13^C-Labeled Escherichia coli

^13^C-labeled *E. coli* was prepared as described previously [32] with some modifications. *E. coli* strain OP50 were cultivated at 37 °C in Luria-Bertani (LB) medium containing 10 μg mL^−1^ streptomycin for 8 h. As the next step, 100 µL of cultured cells were transferred to a flask containing 12.5 mL of M9 minimal medium (42.3 mM Na_2_HPO_4_, 22.0 mM KH_2_HPO_4_, 8.56 mM NaCl, and 18.7 mM NH_4_Cl,) with 1 mM MgSO_4_, 50 μM CaCl_2_, 4 mg L^−1^ thiamine hydrochloride, 5 mg L^−1^ cholesterol, and 2 g L^−1^ D-Glc (1-^13^C), D-Glc (U-^13^C_6_), or D-Glc. The culture was incubated at 37 °C for approximately 14 h, then the cultures were transferred into a flask with 497.5 mL of D-Glc (1-^13^C), D-Glc (U-^13^C_6_), or D-Glc-containing M9 medium, and cultured for 9 h. Cells were harvested by centrifugation at 6000× *g* for 5 min. The cell pellet was washed with M9 medium, then resuspended in 1.5 mL of the same medium. Finally, 150 µL of resuspended cells were seeded onto NGM plates. Worms were transferred to each of the 10 NGM plates and cultured at 20 °C for 85 h.

### 4.4. Preparation of ^13^C-Labeled Spinach Leaves

D-Man was reportedly converted into AsA more effectively than D-Glc in *Arabidopsis thaliana* [20]. Therefore, in order to produce L-AsA (1-^13^C), we provided spinach leaves with D-Man (1-^13^C). Spinach was purchased from a local market in Tottori City, Japan. Detached leaves were provided with 10 mM D-Man or D-Man (1-^13^C) solution under continuous illumination at 24 μmol m^−2^ s^−1^ at 26 °C for 24 h. The treated leaves were kept frozen at −20 °C.

### 4.5. GC-MS Analysis

The GC-MS analysis was carried out as described previously [32] with some modifications. Worms were washed with ice-cold 100 mM potassium phosphate buffer (pH 7.6) into a 50 mL plastic tube and centrifuged at 500× *g* for 1 min. The supernatant was removed and the worms were washed four times with 100 mM potassium phosphate buffer (pH 7.6). The worms were then transferred into microcentrifuge tubes containing 300 µL of 0.5 mM butylated hydroxytoluene-supplemented methanol. Next, sea sand was added into the tube at a volume approximately equivalent of 100 µL and the samples were ground on ice using a disposable plastic pestle. The D-Man or D-Man (1-^13^C)-supplemented spinach leaves were ground into a fine powder in liquid nitrogen using a mortar and pestle. The powdered sample was then homogenized thoroughly in 0.5 mM butylated hydroxytoluene-supplemented methanol. The homogenate was centrifuged at 100,000× *g* for 10 min at 4 °C. The supernatant was transferred into a glass vial and the methanol was evaporated in a centrifugal concentrator (Integrated Speed Vac System ISS110; Savant Instruments Inc., NY, USA). Next, 100 µL of toluene was added into the vials and the samples were dried under N_2_ gas to remove residual water. Finally, the samples were derivatized in 50 µL of BSTFA/TMCS mixture (99:1, *v*/*v*) at 60 °C for 90 min.

GC-MS analysis was performed using a GCMS-QP2010 Plus system (Shimadzu Co., Kyoto, Japan) with a fused silica capillary column (Rtx-5MS; inner diameter, 30 mm × 0.25 mm; film thickness, 0.25 µm; Restek Co., Bellefonte, PA, USA). Helium was used as a carrier gas at a flow rate of 0.81 mL min^−1^. The oven temperature was set as follows: holding at 140 °C for 1 min; heating at 15 °C min^−1^ until 300 °C; holding at this temperature for 6 min. The injector temperature was maintained at 250 °C. The ion source and interface temperatures were 230 °C and 180 °C, respectively. Sample injections (1 μL) were performed in split mode (1:50). The MS detection parameters were set as follows: interface and source temperatures 180 °C and 230 °C, respectively; mass range, *m*/*z* 40–500; scan speed: 1666 amu s^−1^; event time: 0.30 s. Data collection and handling was performed using the GCMS solution (ver. 2.53SU3, Shimadzu) software. We used the National Institute of Standards and Technology (NIST) library in this study.

### 4.6. RNAi Vector Construction

Total RNA (10 μg) of mixed staged-worms was isolated using Sepasol-RNA I (Nacalai Tesque, Kyoto, Japan). Complementary DNA (cDNA) was synthesized using PrimeScript Reverse Transcriptase (Takara Bio, Shiga, Japan) according to the manufacturer’s instructions. Each cDNA fragment from putative AsA biosynthesis-related genes was amplified from *C. elegans* cDNA by PCR and cloned into the pPD129.36 RNAi vector. The primer sequences used for RNAi vector construction were as follows: E03H4.3-F (5′-GCTCTTCTTACGTTGGCGAC-3′), E03H4.3-R (5′-AACCGGTGAAGAGAGACGAA-3′), Y50D7A.7.-F (5′-ATGTCGGCGTCCTATCAAAC-3′), and Y50D7A.7.-R (5′-GTGACTCCCTGAGCTTTTGC-3′). DNA sequencing was commissioned to Eurofins Genomics (Tokyo, Japan).

### 4.7. RNAi Experiments

We used the RNAi hypersensitive KP3948 *eri-1(mg366);lin-15b(n744)* strain [54] for the experiments. RNAi was induced by feeding, as described previously [55], with modifications. Worms were fed the HT115(DE3) *E. coli* strain transformed with the empty pPD129.36 RNAi vector (control), or its appropriate insert-carrying version. Transformed *E. coli* were grown in 5 mL of 2 × YT medium supplemented with ampicillin (50 µg mlL^−1^) and tetracycline (12.5 µg mL^−1^) at 37 °C. When the culture reached an absorbance of 0.4 at 600 nm, 0.5 mM of isopropyl β-D-thiogalactopyranoside (IPTG) was added, and the bacteria were grown for a further 4 h at 37 °C. Finally, 240 µL of cell culture was seeded onto NGM plates containing 0.5 mM IPTG, 50 µg/mL ampicillin, and 12.5 µg/mL tetracycline. Gravid worms were placed onto these RNAi plates and were grown at 20 °C for 3 days.

### 4.8. AsA Measurement

AsA was assayed as described previously [56]. AsA was oxidized into dehydro-AsA, derivatized into its osazone, then assayed using a high-performance liquid chromatography (HPLC) system: a Shimadzu HPLC apparatus, comprising LP-6A pumps, an SPD-6AV UV-visible detector, a CTO-6V column oven, and a CDS ver. 5 chromato-data processing system (LAsoft, Ltd., Chiba, Japan). Each sample (20 μL) was applied on a normal-phase HPLC column (Senshu pak, φ 6.0 × 150 mm) and eluted with acetic acid/hexane/ethyl acetate (1:4:5, *v*/*v*/*v*) at 40 °C. The osazone derivatives were monitored by measuring their absorbance at 495 nm. The flow rate was 1.5 mL min^−1^.

### 4.9. C. elegans Supplementation with AsA Precursor L-Gulono-1,4-lactone or L-Galactono-1,4-lactone

*C. elegans* reportedly incorporates efficiently artificial liposome-packaged nutrients [57]. For stock solution preparation, lipids were dissolved in chloroform and stored at −30 °C. The mixture of 500 nmol DOPC and 50 nmol cholesterol was added into glass vials, then the chloroform was evaporated under a gentle nitrogen gas stream upon continuous mixing to form a thin lipid film on the wall of the vials. These vials were then dried overnight using a SpeedVac concentrator (System ISS110; Savant Instruments Inc., NY, USA). For precursor encapsulation into liposomes, 500 µL of 1 M L-gulono-1,4-lactone, 1 M L-galactono-1,4-lactone, or water (control), were added into a dry lipid vial. The mixture was then sonicated. The prepared liposomes were purged with N_2_ gas and stored at −80 °C.

### 4.10. qPCR Analysis

Total RNA was used to synthesize cDNA using the PrimeScript™ RT Reagent Kit with gDNA Eraser (Takara Bio, Shiga, Japan). Primer sets were designed using GENETYX-MAC (GENETYX Corporation, Tokyo, Japan) as follows: F54D5.11-F (5′-TGTCGGATTTGGCTGAGTGT-3′), F54D5.11-R (5′-ACTCCGTCCATATGCTCGTT-3′), Act-1-F (5′-TCCAAGAGAGGTATCCTTACCC-3′), and Act-1-R (5′-CTCCATATCATCCCAGTTGGTG-3′). A CFX Connect Real-Time System (Bio-Rad Laboratories, Inc., Hercules) with GeneAce SYBR^®^ qPCR Mix α (Nippon Gene Co., Ltd., Tokyo, Japan) was used to perform qPCR.

### 4.11. Collagen to Non-Collagen Protein Ratio Measurement

The ratio of collagen to non-collagen protein was determined using a Sirius Red/Fast Green Collagen Staining Kit (Chondrex, Inc., Redmond, WA, USA), following the manufacturer’s instructions.

### 4.12. C. elegans Lifespan Determination

N2 and tm6671 worms were allowed to lay eggs for 8–12 h to obtained synchronized worms. The next day, L1 worms were then transferred to 30 mm NGM plates seeded with OP50 (~12 worms each) and maintained at 20 °C. Worms were transferred every 2 days to new plates after reaching adult stages to avoid confusion between generations. Day 1 of the lifespan was started after egg laying ended and scoring was performed every day until the last worm died. Worms were considered dead when they no longer responded to mechanical stimulation. Worms lost during the observation and those with internally hatched larvae were excluded from the experiment. Approximately 250 worms were used in the lifespan experiment.

### 4.13. Malondialdehyde Measurement

Malondialdehyde (MDA) levels were determined using a TBARS assay kit (ZeptoMetrix Crop., Buffalo, NY, USA), following the manufacturer’s instructions. We determined MDA-thiobarbituric acid adducts formation from the MDA reaction in the samples measuring the absorbance at 540 nm using a microplate reader (Sunrise Rainbow RC-R, TECAN, Männedorf, Switzerland).

### 4.14. Statistical Analysis

We used GraphPad Prism 3 for Windows, version 2.01 (GraphPad Software Inc., La Jolla, CA, USA) for statistical analysis. AsA and MDA levels and collagen to non-collagen protein ratios were evaluated using one-way ANOVA with Tukey’s multiple comparison test. All data, except for the lifespan, are presented as the mean ± standard deviation (SD). Significant differences were defined as *p* < 0.05. We used the Log rank test for survival curve comparison with *p*-values < 0.001 defined as significant. Kaplan–Meier survival curves are shown in the lifespan figures.

## Figures and Tables

**Figure 1 metabolites-10-00334-f001:**
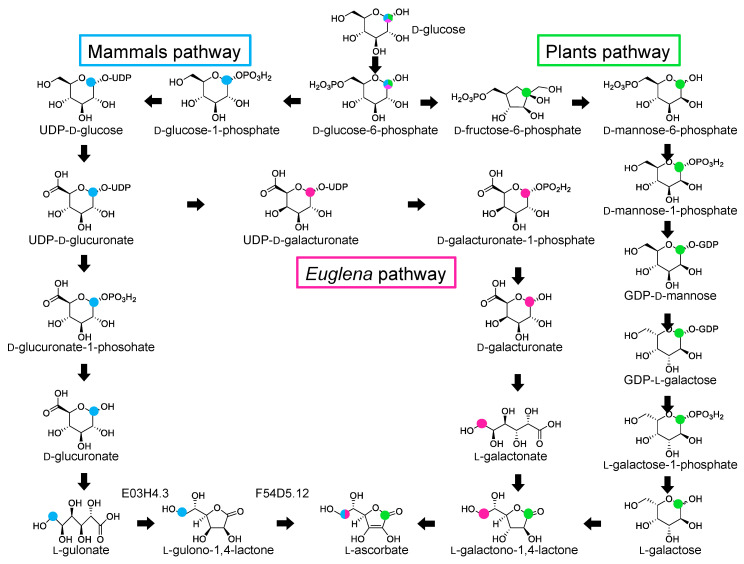
Ascorbate (AsA) biosynthetic pathways in mammals, plants, and *Euglena*. Overview of the major ascorbate pathways [20,21,28]. The mammalian and *Euglena* pathways involve carbon skeleton rearrangement, whereas the plant pathway does not.

**Figure 2 metabolites-10-00334-f002:**
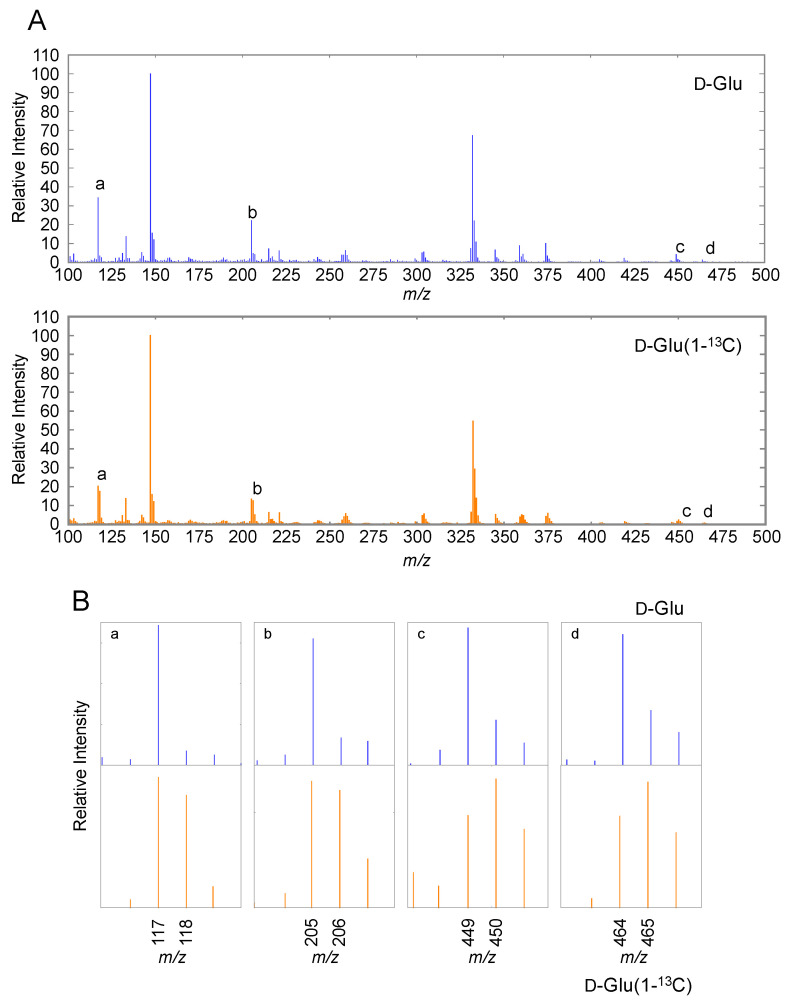
Analysis of AsA in D-Glc or D-Glc (1-^13^C)-labeled *Escherichia coli*-fed *Caenorhabditis elegans* by gas chromatography-mass spectrometry (GC-MS). *C. elegans* preparation and GC-MS analysis were performed as described in Materials and Methods. GC-MS fragment spectrum of the *C. elegans* extract eluted at 8.1 min (**A**). A BSTFA/TMCS-derivatized extract of non-labeled or D-Glc (1-^13^C)-labeled *E. coli*-fed *C. elegans* (**B**). An enlarged view of the *m*/*z* for the specific fragments labeled a–d is shown in the upper panels. Blue line indicates GC-MS fragment pattern in non-labeled *E. coli*-fed *C. elegans* (D-Glc). Orange line indicates D-Glc (1-^13^C)-labeled *E. coli*-fed *C. elegans* (D-Glc (1-^13^C)).

**Figure 3 metabolites-10-00334-f003:**
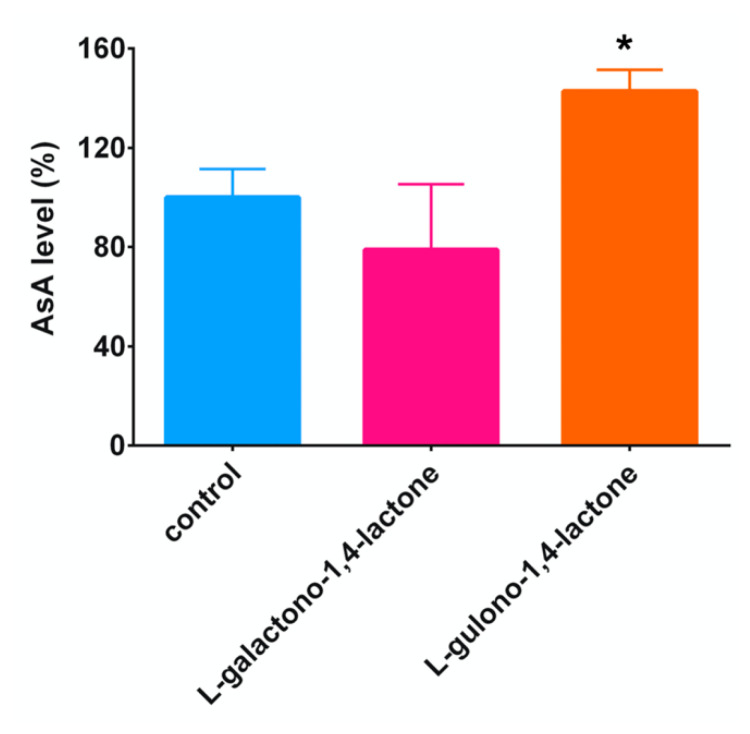
Effect of the addition of L-gulono-1,4-lacton or L-galactono-1,4-lacton on AsA level in worms. AsA level was measured as described in Materials and Methods. Data are expressed as percentages of the values in the liposome-fed worms, used as a control. All values represent mean ± SD of three independent experiments. Asterisks indicate that the values are significantly different from those in the control worms (*p* < 0.05).

**Figure 4 metabolites-10-00334-f004:**
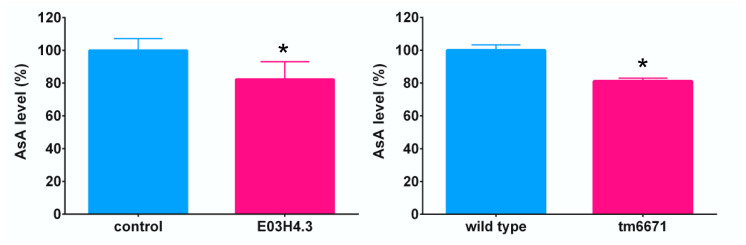
Effect of knockdown of E03H4.3 expression or knockout of F54D5.12 on AsA level in worms. AsA level was measured as described in Materials and Methods. Data are expressed as percentages of the values in the control or N2 worms. All values represent mean ± SD of three independent experiments. Asterisks indicate that the values are significantly different from those in the control or N2 worms (*p* < 0.05).

**Figure 5 metabolites-10-00334-f005:**
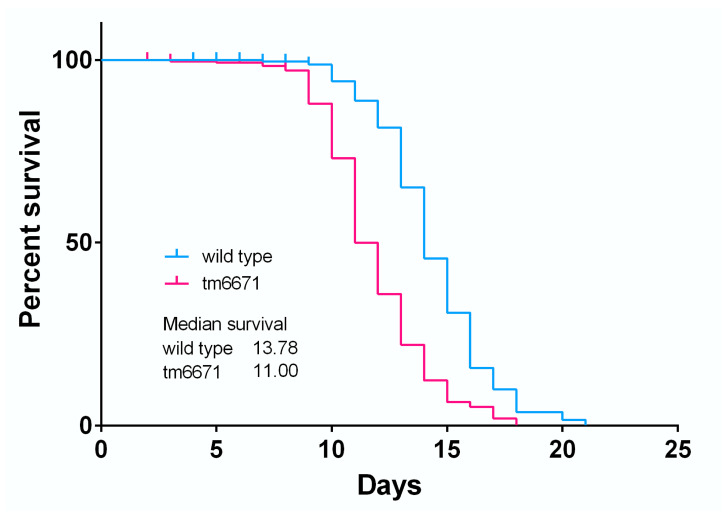
Kaplan–Meier survival curves of N2 and tm6671 mutant worms. Survival curves are shown for N2 and tm6671 mutant worms. The live/dead nematodes were scored every 24 h until all dead. Experimental conditions are described in Materials and Methods. The graph is representative of three independent biological replicates. Median lifespan, days: N2, 13.78; tm6671, 11.00. The log-rank test *p*-value for each experiment was < 0.001.

**Figure 6 metabolites-10-00334-f006:**
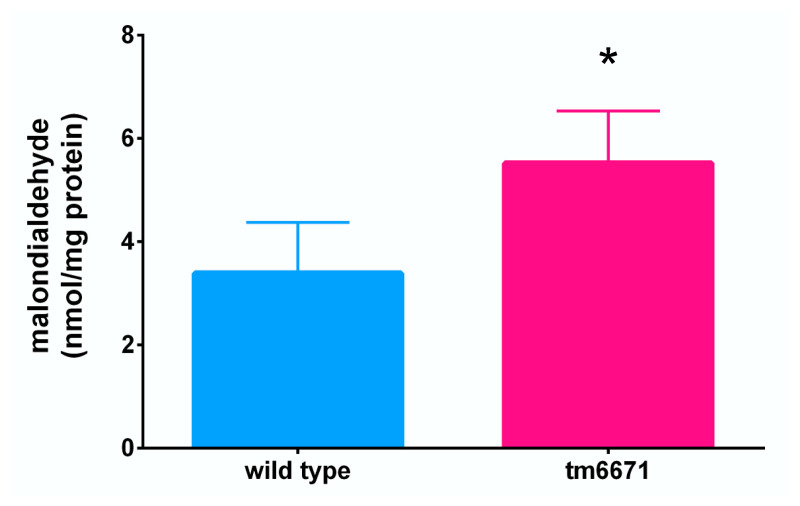
MDA level in the tm6671 mutant worms. MDA level was measured as described in Materials and Methods. All values represent mean ± SD of seven independent experiments. Asterisks indicate that the values are significantly different from those of the N2 worms (*p* < 0.05).

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
