# Peer review of "L-Ascorbate Biosynthesis Involves Carbon Skeleton Rearrangement in the Nematode Caenorhabditis elegans"

_metabolites, 2020, doi:10.3390/metabo10080334_

Round 1

Reviewer 1 Report

Yabuta et al present an analysis of vitamin C biosynthesis in C. elegans.

They demonstrate that

  • Ascorbate AsA biosynthesis in C. elegans involves carbon skeletal rearrangement – this work is robust and clearly presented.
  • C. elegans use a L-gulono but not a L-galactono lactone precursor for biosynthesis of AsA - clearly presented and robust

They appropriately conclude that the pathway for AsA synthesis in C. elegans is likely similar to the pathway found in mammals and distinct from biosynthetic pathway in plants or Euglena.

They identify genes encoding enzymes that likely contribute to AsA biosynthesis and demonstrate enzymatic activity of the E03H4.3 gluconolactonase.

Major question: I have a lingering question about the activity / specificity of the “knockout” allele tm6671. A brief description of the nature of the allele should be provided. It appears to be a small deletion that removes part of the gene and a portion of the 3’ end of a second gene F54D5.11. How is this allele predicted to affect the coded protein? Knockdown of the second gene F54D5.11 should be investigated as a control for the AsA measurements, even more importantly for the life span and oxidative stress experiments, which would likely be sensitive to perturbation of many other genes.

Finally, they show that reduction of function of F54D5.12(tm6671) results in a reduced life span and evidence of oxidative stress but no change in morphology or global collagen protein levels.

Major Question: The images used to demonstrate a lack of morphological difference between (Supp Fig. 5) the N2 and F54D5.12(tm6671) are not convincing. In fact tm6671 look fatter, less mobile (no body bend), and more full of eggs. Do these animals retain eggs that hatch internally?  If so, did this affect life span assessments? More methods details on life span assay would be helpful – for example, were any animals censured?

Minor questions and comments

N2 is not a wild type strain and should be referred to as a lab type or control.

 https://www.ncbi.nlm.nih.gov/pmc/articles/PMC4417040/

The language in the paragraph starting on line 163 is misleading, suggesting that RNAi of the candidate genes decreased AsA levels by 80% as opposed to decreasing them to 80% of normal.  

Have the authors sought to have E03H4.3 renamed (via appropriate C. elegans community protocols) to reflect gene activity?

Findings related to Y7A5A.1 are mentioned in the discussion (line 223) but not presented in the results.  

In Supp Fig 4 the standard deviations for the AsA level measurements are much higher than in Fig 4. This raises questions about the real difference between the 20% decrease seen in Fig 4 (statistically significant) and the ~20% decrease seen in Supp Fig 4 (not statistically significant). Also, the data presented is said to be normalized to the control or the “wild type”, yet the control value plotted is not at 100%. It seems that something is off here.

Minor grammar issues:

Line 185

Lines 235-6

Author Response

Responses to comments of reviewer #1:

First, we would like to decline ‘’2.4. The recombinant E03H4.3 protein exhibits gluconolactonase activity’’.  When we checked gluconolactonase activity again, we perceived that slightly contaminating protein exhibits gluconolactonase activity. We are sorry. Therefore, we delated the descriptions related to gluconolactonase activity in the revised MS.  However, the suppression of E03H4.3 decreased AsA level in worms.

Thank you very much for your assistance in reviewing our MS (metabolites-879012).  We appreciate your helpful suggestions and comments. We have tried to revise the manuscript in line with suggestions.

They demonstrate that Ascorbate AsA biosynthesis in C. elegans involves carbon skeletal rearrangement – this work is robust and clearly presented. C. elegans use a L-gulono but not a L-galactono lactone precursor for biosynthesis of AsA - clearly presented and robust They appropriately conclude that the pathway for AsA synthesis in C. elegans is likely similar to the pathway found in mammals and distinct from biosynthetic pathway in plants or Euglena. They identify genes encoding enzymes that likely contribute to AsA biosynthesis and demonstrate enzymatic activity of the E03H4.3 gluconolactonase.

Major question:

Q1    I have a lingering question about the activity / specificity of the “knockout” allele tm6671. A brief description of the nature of the allele should be provided. It appears to be a small deletion that removes part of the gene and a portion of the 3’ end of a second gene F54D5.11. How is this allele predicted to affect the coded protein?  Knockdown of the second gene F54D5.11 should be investigated as a control for the AsA measurements, even more importantly for the life span and oxidative stress experiments, which would likely be sensitive to perturbation of many other genes.

A1    The 54D5.12 knockout mutants did not exist except for tm6671.  Two types of mRNA are transcribed from F54D5.11 gene. One is F54D5.11.1, another is F54D5.11.2.  Both mRNAs code identical protein, although F54D5.11.2 had long 3’-UTR.  Because the 3’ end of 3’-UTR of F54D5.11.2, but not F54D5.11.1 was delated in tm6671, we thought that the effect on the expression of F54D5.1 protein in tm6671 mutant worms may be small.  So, we chose this mutant.  We understand your suggestion. However, we cannot prepare and perform the knockdown experiment of F54D5.11, within for 10 days. We are sorry.

Finally, they show that reduction of function of F54D5.12(tm6671) results in a reduced life span and evidence of oxidative stress but no change in morphology or global collagen protein levels.

Major Question:

Q2.   The images used to demonstrate a lack of morphological difference between (Supp Fig. 5) the N2 and F54D5.12(tm6671) are not convincing. In fact tm6671 look fatter, less mobile (no body bend), and more full of eggs. Do these animals retain eggs that hatch internally? If so, did this affect life span assessments? More methods details on life span assay would be helpful – for example, were any animals censured?

A2. Although we took new pictures, we could not observe significant difference in morphology between the N2 and F54D5.12 knockout mutant worms.  We found only one tm6671 worm with internally hatched larvae during lifespan assay.  Worms lost during the observation and those with internally hatched larvae were excluded from the experiment.  These pictures were shown in new Supplementary Figure S7 in the revised MS.  According your comment, we added more methods details on life span assay. (revised MS page 10, lines 521-528).

Minor questions and comments

Q3.   N2 is not a wild type strain and should be referred to as a lab type or control. https://www.ncbi.nlm.nih.gov/pmc/articles/PMC4417040/

A3.   According to reviewer’s comment, we changed from wild-type to N2 in the revised MS. Furthermore, we changed Figs. 4, 5 and 6 in the revised MS.

Q4.   The language in the paragraph starting on line 163 is misleading, suggesting that RNAi of the candidate genes decreased AsA levels by 80% as opposed to decreasing them to 80% of normal.

A4.   Thank you for finding our mistake.  we changed this sentence in the revised MS. (revised MS page 5, lines 169-170).

Q5.   Have the authors sought to have E03H4.3 renamed (via appropriate C. elegans community protocols) to reflect gene activity?

A5.   Described above, we did not detected gluconolactonase activity E03H4.3 protein.  After we perform a more detailed analysis, and the gene name will be renamed.

Q6.   Findings related to Y7A5A.1 are mentioned in the discussion (line 223) but not presented in the results.

A6.   We are sorry to mistake.  We had intended to write about a future plan. We corrected to “we are going to analyze …”. (revised MS page 7, lines 280-281).

Q7.   In Supp Fig 4 the standard deviations for the AsA level measurements are much higher than in Fig 4. This raises questions about the real difference between the 20% decrease seen in Fig 4 (statistically significant) and the ~20% decrease seen in Supp Fig 4 (not statistically significant). Also, the data presented is said to be normalized to the control or the “wild type”, yet the control value plotted is not at 100%. It seems that something is off here.

A7.   We are sorry our mistake. Left graph of Supplemental Fig. 4. was corrected in the revised MS.

Since AsA is unstable compound, we are careful experimental conditions.  AsA level of F54D5.12 knockout worms or E03H4.3 suppressed worms were lower than that in the N2 or control worms at every time.  In contrast, the suppression of Y50D7A.7 expression or mutation of Y50D7A.7 did not affect to AsA level.

Therefore, we judged that the suppression of E03H4.3 or the deficiency of F54D5.12 significantly decreased AsA level in C. elegans and that Y50D7A.7 dose not involve in AsA biosynthesis. 

Q8.   Minor grammar issues:

Line 185

L0ines 235-6

A8.        According to reviewer’s comment, we corrected them.

Reviewer 2 Report

Summary: Vitamin C, also known as L-Ascorbate (AsA), is an important biological reductant and serves as a cofactor in diverse biological processes including the synthesis of collagen, neuromodulation and oxidative metabolism. Lack of AsA causes a disease called scurvy, a condition manifested as bleeding abnormalities due to due to impaired collagen synthesis and disordered connective tissue. Plans and most animals can synthesize AsA, except for the primates and guinea pigs. Yabuta and colleagues investigate the AsA synthesis in the invertebrate nematode Caenorhabditis elegans. A previous study has demonstrated that ascorbate is present and synthesized by the nematode, however, its biosynthetic and metabolic pathway are unknown. In this study, by using GC-MS analysis and feeding worms with AsA precursors, the authors show that, AsA synthesis in C. elegans is similar to the mammalian pathway and it involves carbon skeleton rearrangement. Furthermore, the authors identify E03H4.3 and F54D5.12 as the potential worm orthologs of the AsA biosynthetic enzymes SMP30/GNL and GULO, respectively. Only knockout of the F54D5.12 gene is found to reduce the AsA level in the worms. Interestingly, while F54D5.12 knockout mutants show no significant change in collagen content and normal morphology, they show an elevation of MDA suggesting increased oxidative stress, and their lifespan is shortened. Overall, the paper is easy to follow, and the data is clearly presented. However, the following revisions should be considered:

  • Sequence alignments between the mammalian SMP30/GNL or GULO and the identified worm orthologs should be shown.
  • The authors show that F54D5.12 mutant worms with reduced AsA levels and an elevation of the oxidative stress marker MDA live shorter than wild-type worms. AsA supplementation will address whether the lifespan effect is a consequence of lower AsA level or not, and therefore should be performed and presented. Causal relationships between AsA levels and the effects on oxidative stress and lifespan are convincing only if the authors could demonstrate that restoration of AsA levels in F54D5.12 mutants rescues the other phenotypes. Otherwise, I suggest the authors consider re-formatting the related conclusions or statements: for example, lines 204-205: “These data demonstrate for the first time that AsA deficiency significantly reduces C. elegans lifespan”.
  • Transgene rescue experiment for 12(tm6671) would be necessary to confirm that F54D5.12 is required for AsA production.
  • Have the authors examined other oxidative stress markers in addition to MDA in F54D5.12 mutant?
  • Line 224-225: “To examine the involvement of these proteins in AsA biosynthesis, we analyzed how the reduced Y7A5A.1 or F52H2.6 expression levels affect AsA level”. But no data presented that I could find.
  • The discussion section should be revised substantially. Firstly, sections that are not directly relevant to the manuscript’s findings, for example, the discussion about SVCT2 (lines 260-277), should be removed or cut down significantly. Second, the discussion on the possible link between decreased AsA level and stress response pathways (lines 240-247) is too superficial and speculative. I suggest removing it.

Author Response

Responses to comments of reviewer #2:

First, we would like to decline ‘’2.4. The recombinant E03H4.3 protein exhibits gluconolactonase activity’’.  When we checked gluconolactonase activity again, we perceived that slightly contaminating protein exhibits gluconolactonase activity. We are sorry. Therefore, we delated the descriptions related to gluconolactonase activity in the revised MS.  However, the suppression of E03H4.3 decreased AsA level in worms.

Thank you very much for your assistance in reviewing our MS (metabolites-879012).  We appreciate your helpful suggestions and comments. We have tried to revise the manuscript in line with suggestions.

Reviewer 2

Comments and Suggestions for Authors

Summary: Vitamin C, also known as L-Ascorbate (AsA), is an important biological reductant and serves as a cofactor in diverse biological processes including the synthesis of collagen, neuromodulation and oxidative metabolism. Lack of AsA causes a disease called scurvy, a condition manifested as bleeding abnormalities due to due to impaired collagen synthesis and disordered connective tissue. Plans and most animals can synthesize AsA, except for the primates and guinea pigs. Yabuta and colleagues investigate the AsA synthesis in the invertebrate nematode Caenorhabditis elegans. A previous study has demonstrated that ascorbate is present and synthesized by the nematode, however, its biosynthetic and metabolic pathway are unknown. In this study, by using GC-MS analysis and feeding worms with AsA precursors, the authors show that, AsA synthesis in C. elegans is similar to the mammalian pathway and it involves carbon skeleton rearrangement. Furthermore, the authors identify E03H4.3 and F54D5.12 as the potential worm orthologs of the AsA biosynthetic enzymes SMP30/GNL and GULO, respectively. Only knockout of the F54D5.12 gene is found to reduce the AsA level in the worms. Interestingly, while F54D5.12 knockout mutants show no significant change in collagen content and normal morphology, they show an elevation of MDA suggesting increased oxidative stress, and their lifespan is shortened. Overall, the paper is easy to follow, and the data is clearly presented. However, the following revisions should be considered:

Q1.   Sequence alignments between the mammalian SMP30/GNL or GULO and the identified worm orthologs should be shown.

A1.   According to reviewer’s comment, we added sequence alignments as New Supplemental Figures S4 and 5 in the revised MS.

Q2.   The authors show that F54D5.12 mutant worms with reduced AsA levels and an elevation of the oxidative stress marker MDA live shorter than wild-type worms. AsA supplementation will address whether the lifespan effect is a consequence of lower AsA level or not, and therefore should be performed and presented. Causal relationships between AsA levels and the effects on oxidative stress and lifespan are convincing only if the authors could demonstrate that restoration of AsA levels in F54D5.12 mutants rescues the other phenotypes. Otherwise, I suggest the authors consider re-formatting the related conclusions or statements: for example, lines 204-205: “These data demonstrate for the first time that AsA deficiency significantly reduces C. elegans lifespan”.

A3.   We understand the importance of such rescue experiments.  Now, we are studying rescue experiments using the stable AsA derivatives.  However, we do not have enough data to show in our MS. According to your comment, we delated the related conclusion in result and discussion sections in new MS. 

Q3.   Transgene rescue experiment for 12(tm6671) would be necessary to confirm that F54D5.12 is required for AsA production.

A3.   Thank you very much for your fruitful suggestion.  However, we did not attempt transgene rescue experiment.  We are sorry, it was difficult to perform such experiment within for 10 days.  In forthcoming paper, we would like to discuss it.

Q4.   Have the authors examined other oxidative stress markers in addition to MDA in F54D5.12 mutant?

A4.   Although we attempted to observe the ROS accumulation in worms using DCFDA, the variation was big.  Because we did not obtain the exact data, we can not show such data.  

Q5.   Line 224-225: “To examine the involvement of these proteins in AsA biosynthesis, we analyzed how the reduced Y7A5A.1 or F52H2.6 expression levels affect AsA level”. But no data presented that I could find.

A5.   We are sorry to mistake.  We had intended to write about a future plan. We corrected to “we are going to analyze …”. . (revised MS page 7, lines 280-281).

Q6.   The discussion section should be revised substantially. Firstly, sections that are not directly relevant to the manuscript’s findings, for example, the discussion about SVCT2 (lines 260-277), should be removed or cut down significantly. Second, the discussion on the possible link between decreased AsA level and stress response pathways (lines 240-247) is too superficial and speculative. I suggest removing it.

A6.   According to reviewer’s comment, we delated two sections it the revised MS.

Round 2

Reviewer 1 Report

Minor concerns with the manuscript were addressed by the authors. However, the major problem has not been remedied.

The reliance on an allele, tm6671, which has potential effects on two genes, F54D5.12 and F54D5.11, to draw conclusions about F54D5.12 is problematic, and the authors have not addressed this issue.  The allele is not adequately described in the manuscript and no caveats are presented regarding potential complications of the genetic results.

In the absence of additional experiments such as confirming the tm 6671 allele phenotype with RNAi of F54D5.12 (and not with RNAi of F54D5.11), reversion of the phenotype via transgenic expression of F54D5.12 (and not F54D5.11) or reversion of the life span via supplementation with AsA as suggested by the other reviewer, the conclusions drawn regarding life span and AsA levels are too strong/ not fully supported by the evidence.

Author Response

Responses to comments of reviewer #1:

Thank you very much for your assistance in reviewing our MS (metabolites-879012).  We appreciate your helpful suggestions and comments. We have tried to revise the manuscript in line with suggestions. The corrected parts are shown in red letters.

Minor concerns with the manuscript were addressed by the authors. However, the major problem has not been remedied.

Q1.   The reliance on an allele, tm6671, which has potential effects on two genes, F54D5.12 and F54D5.11, to draw conclusions about F54D5.12 is problematic, and the authors have not addressed this issue.  The allele is not adequately described in the manuscript and no caveats are presented regarding potential complications of the genetic results.

A1.   According to your comments, we described that the possibility of small deletion of F54D5.11 gene affects to the F54D5.11 expression in the tm6671 mutant worms in the results section of revised MS (page 5, lines 165-168). Furthermore, we checked the transcript level of F54D5.11 in the N2 and tm6671 mutant worms. No significant difference was observed in the transcript level of F54D5.11 between the N2 and tm6671 mutant worms. It seems likely that the expression of F54D5.11 protein is not affected by tm6671 allele. These finding were added in the revised MS (revised MS page 5, lines 168-171). We also added the method about qPCR analysis. Furthermore, we thought that the "F54D5.12 knockout mutant" was inappropriate, so we changed to "tm6671 mutant”.

Q2.   In the absence of additional experiments such as confirming the tm 6671 allele phenotype with RNAi of F54D5.12 (and not with RNAi of F54D5.11), reversion of the phenotype via transgenic expression of F54D5.12 (and not F54D5.11) or reversion of the life span via supplementation with AsA as suggested by the other reviewer, the conclusions drawn regarding life span and AsA levels are too strong/ not fully supported by the evidence.

A2.   According to your comment, we deleted the conclusions of the relationship between AsA level and lifespan in the revised MS.

Reviewer 2 Report

none

Author Response

Responses to comments of reviewer #2:

Thank you very much for your assistance in reviewing our MS (metabolites-879012).  

Round 3

Reviewer 1 Report

My concerns are mostly mitigated.

In line 166 the authors say it is likely that F54D5.11 expression is affected and then go on to show that it isn't. Thus, I recommend that line 166 be revised to say we considered the possibility that F54D5.11 expression...

Author Response

Responses to comments of reviewer #1:

Thank you very much for your assistance in reviewing our MS (metabolites-879012).  We appreciate your helpful suggestions and comments. According to your comment, we corrected lines 166-167.

Page 5, lines 166-167: We considered the possibility that the transcript level of F54D5.11 was changed in the tm6671 mutant worms.